# Financing sustainability: Applying the BIOFIN framework to government investments in conserving native and indigenous livestock breeds in central India

Bakul Lad[1,2], Faiyaz A. Khudsar[3], Ajay Sharma [4], Randeep Singh [1]*

**1** Amity Institute of Forestry and Wildlife, Amity University Uttar Pradesh, Noida, India, **2** Madhya Pradesh Biodiversity Board, Bhopal, Madhya Pradesh, India, **3** Biodiversity Parks Programme, Centre for Environmental Management of Degraded Ecosystems, University of Delhi, Delhi, India, **4** College of Forestry, Wildlife and Environment, Auburn University, Auburn, Alabama, United States of America

\* rsingh18@amity.edu

## Abstract

Biodiversity conservation is a key responsibility for signatories of the Convention on Biodiversity. India is legally committed to the Aichi Biodiversity Targets (ABTs) and the United Nations' sustainable development goals (SDGs), with a focus on protecting its livestock diversity and genetic resources. Livestock diversity is crucial for maintaining ecosystem functionality and supporting the livelihoods of indigenous communities by contributing to the rural economy. However, urbanization, rapid land use changes, agricultural mechanization, loss of pastoral lands, and emerging diseases have significantly reduced indigenous livestock diversity. The growing demand for high-yield livestock breeds further threatens native and indigenous varieties. This study examines the financial mechanisms employed by the public sector Animal Husbandry Department in Madhya Pradesh, India, using the Biodiversity Finance Initiative (BIOFIN) framework. It assesses the availability of funding across five BIOFIN categories—conservation, sustainable use, education and awareness, policy, and access and benefit sharing—to meet state biodiversity targets aligned with the Aichi Targets. A total of 43 schemes were identified and mapped to these categories. From 2016 to 2022, approximately INR 5,159.88 crore (~US $727.4 million) was invested in livestock conservation and sustainable production in the state. The BIOFIN methodology has facilitated robust financial planning for the livestock sector. The sector is meeting the financial needs assessed in the Madhya Pradesh Biodiversity Strategy and Action Plan, 2018−30, through systematic investments for the conservation and sustainability of livestock biodiversity with an effective awareness extension program. While progress has been made, the emphasis on increasing native livestock production continues to pose challenges to biodiversity conservation.

**Data availability statement:** All relevant data are within the manuscript and its Supporting Information files.

**Funding:** The author(s) received no specific funding for this work.

**Competing interests:** The authors have declared that no competing interests exist.

## Introduction

Funding for sustainability is often overlooked, contributing to biodiversity loss and exacerbating global environmental and social issues [1]. However, finance for sustainability is essential for addressing these issues [2]. It involves integrating biodiversity considerations into Environmental, Social, and Governance (ESG) criteria within financial decision-making processes both for retail investors and public sector authorities to enhance long-term sustainability [3–5].

The Sustainable Development Goals (SDGs) emphasize the significance of protecting ecosystems and biodiversity [6,7]. To achieve the ambitious 2030 targets of the sustainable development agenda, the United Nations Development Programme (UNDP) has called for increased SDGs financing through innovative approaches, policy decisions, and new funding sources [8]. According to Arlaud et al. (2018), global biodiversity conservation efforts require an estimated annual investment of $150 billion to $440 billion. To address the financial challenges associated with biodiversity conservation, the UNDP introduced the Biodiversity Finance Initiative (BIOFIN) in 2012 [9]. This initiative aims to create a method for measuring the biodiversity finance gap at the national level and improve cost-effectiveness by integrating biodiversity into national development and sectoral planning [9]. BIOFIN assists countries in optimizing their investments in biodiversity, ensuring that financial resources are effectively allocated to meet conservation goals.

The BIOFIN approach is currently being piloted in over 40 countries, including 10 of the world's 17 most biodiverse nations (Brazil, Indonesia, Colombia, China, Madagascar, Malaysia, Mexico, Peru, India, and Ecuador) [10]. It aims to strengthen national biodiversity finance frameworks and close the financing gap for biodiversity conservation and sustainable use [10]. The sustainable use of biodiversity is essential for food security, as it is closely connected to food production and agriculture, making the protection of indigenous species necessary [11]. In India, BIOFIN was launched in 2015 by the Ministry of Environment, Forest and Climate Change (MoEFCC) through the National Biodiversity Authority (NBA). India is among the select nations that have formulated 12 National Biodiversity Targets (NBTs), aligning them with the 20 global Aichi targets and revising its national biodiversity action plan to incorporate these national targets [12]. Its primary goal is to mobilze resources for biodiversity conservation and ensure that both private and public, domestic and international finances align with the National Biodiversity Action Plan (NBAP), the National Biodiversity Targets (NBTs), and related commitments [13]. India is also legally bound to Aichi Biodiversity Targets (ABTs) and the SDGs, and it is actively working to protect its biodiversity and its genetic resources [14]. However, achieving these targets requires significant financial investment to conserve land and water resources and protect the genetic diversity of plants, indigenous livestock, and other valuable species. This is critical to minimize genetic erosion and safeguard biodiversity [15].

Biodiversity not only supports food production but also improves the nutritional sustainability of the food supply, strengthens community resilience, and improves rural livelihoods [16]. In the agricultural economy, livestock plays a crucial role and is

one of the fastest-growing sectors, with an estimated global value of at least $1.4 trillion [17]. It significantly contributes to food and nutrition security and supports the livelihoods of 600 million smallholder farmers [18]. Livestock accounts for over 40% of the food supply in developing countries [19] and is the largest user of land resources globally, occupying 33% of the world's land surface [20].

India is recognised as a megadiverse country with rich biodiversity. Despite facing significant biotic pressure, it supports 7−8% of the world's recorded species, 18% of the global human, and 10% of the global livestock population— all within just 2.4% of the global land area [21]. Livestock plays a vital role in the lives of millions of Indians, contributing to their socio-economic well-being [22]. Beyond its economic value, livestock is culturally significant used in religious ceremonies, as a symbol of social status, and in recreational activities [23]. In the fiscal year 2022−23, the livestock sector contributed over 4.66% to India's Gross Domestic Product (GDP) (National Accounts Statistics 2024). Historically, livestock development in India has focused on introducing exotic breeds and crossbreeding them with local breeds to enhance milk production [24]. However, these exotic breeds are often more susceptible to climate-related challenges like disease and heat stress [25].

Livestock breeds and their genetic diversity are integral components of global biodiversity [26]. India is home to a range of indigenous livestock genetic diversity, encompassing 212 breeds of farm animals and poultry species (cattle, buffalo, sheep, goats, poultry, camels, equines, yaks, and Mithun) distributed across its various agro-climatic regions (NBAGR 2022). India's indigenous livestock breeds are particularly well-adapted to harsh environmental conditions and have shown potential for adaptation in other parts of the world [24]. For example, India has exported several indigenous breeds, such as Ongole, Gir, and Sahiwal, to countries like Brazil, the United States, Argentina, and Mexico to improve livestock germplasm [24,27].

Madhya Pradesh state, located in central India, has a diverse and significant livestock population that plays a critical role in the state's economy and rural livelihoods. The state contributes over 7.5% to India's total livestock population, ranking third nationally with approximately 40.6 million animals [28]. Livestock farming is a key source of income for many rural households in the region [29,30].

The native and indigenous livestock breeds in Madhya Pradesh are well-adapted to the local climate and are known for their resilience and productivity [31]. However, factors such as rapid urbanization, agricultural mechanization, loss of pasture lands, rising demand for high-yield cattle, and disease outbreaks pose significant threats to livestock biodiversity [32]. The genetic makeup of native and indigenous livestock breeds is an important component of biodiversity, and it plays a crucial role in climate resilience, social development, and economic sustainability [33]. In this study, we focused on breeds that are either native to Madhya Pradesh or indigenous to India. While native breeds are specific to the state, indigenous breeds may originate from other parts of the country [34]. The main types of livestock in Madhya Pradesh include cattle (*Bos taurus*), buffaloes (*Bubalus bubalis*), goats (*Capra hircus*), and poultry [34,35]. Notable native cattle breeds include Malvi, Nimari, Galo, and Kenkatha, while indigenous breeds such as Sahiwal, Tharparkar, Hariana, and Gir are also present [23]. The state is home to the native buffalo breed, Bhadawari, and the indigenous buffalo Murrah breed. Goat farming is common, especially in rural areas, with native breeds like Jamunapari and Barbari known for their meat and milk production [31,34]. The Kadaknath chicken (*Gallus gallus domesticus*), a native poultry breed with a Geographical Indication (GI) tag, is also prominent [36].

In the fiscal year 2021–2022, the government of Madhya Pradesh expended a total of INR 74.9 million through various programs and schemes run by the Animal Husbandry Department (S1 Appendix). These efforts were directed towards conserving native and indigenous livestock breeds and improving animal productivity across all species to support livelihoods. The Animal Husbandry Department has assessed the impacts and outcomes of its various initiatives, with a particular focus on disease control, infrastructure development, and the enhancement of animal welfare and productivity. Additionally, the department has advanced breeding programmes and genetic improvement initiatives, complemented by targeted training and awareness campaigns. To improve transparency and understanding of biodiversity-related spending,

the Convention on Biological Diversity (CBD) has encouraged countries to conduct National Biodiversity Expenditure Reviews (NBERs). These reviews help clarify the extent of financial investment in biodiversity conservation.

To meet the NBTs, the state of Madhya Pradesh has developed specific State Biodiversity Conservation Targets (SBTs) for the livestock sector under the Madhya Pradesh State Biodiversity Strategy and Action Plan (MPBSAP), 2018−30. BIOFIN supports the development of Biodiversity Finance Plans through a structured, three-stage process: (1) Policy and Institutional Review (PIR), (2) Biodiversity Expenditure Review (BER), and (3) Financial Needs Assessment (FNA) [9,37]. This study applies the BIOFIN framework to evaluate how public sector investments by the Animal Husbandry Department in Madhya Pradesh contribute to the conservation of native and indigenous livestock. The goal is to align these investments with Aichi Target 13 and National Biodiversity Target 07 (focused on protecting genetic diversity in socio-economically and culturally significant species), as well as Target 4 of the Kunming-Montreal Global Biodiversity Framework (2022), which aims to halt species extinction and protect genetic diversity [38]. Additionally, the study addresses Aichi Target 1 and its corresponding national and global targets related to biodiversity awareness and capacity-building.

## Materials and methods

### Study area

The study was conducted in the Central Indian state of Madhya Pradesh (MP), located at 21°17′ to 26°52′ N latitude and 74°08′ to 82°49′ E longitude, with a total area of 30.8 million km$^2$, which is equivalent to 9.38% of the country's total geographical area [39]. The biodiversity of Madhya Pradesh state includes the diversity of ecosystems, including plateaus, ravines, ridges, valleys, and flat plains belonging to four classified zones: the low-lying areas in the north and north-west of Gwalior, the Malwa Plateau, Satpura, and the Vindhya ranges. The livestock sector in Madhya Pradesh is diverse and dynamic, playing a key role in the state's agricultural landscape and rural economy.

### BIOFIN parameters characterization

We followed BIOFIN guidelines and modified them according to our needs to characterize the parameters used in defining the key elements, as suggested by [40]. The key parameters were (i) outlining the definition of 'biodiversity expenditure' following the guidelines proposed by Ansari et al. (2018) [41] in India. (ii) categorizing various types of biodiversity expenditures that support native and indigenous livestock conservation, as outlined in the BIOFIN 2018 workbook [9] (iii) the expenditures, which include both direct and indirect costs relevant to biodiversity, were aimed at the conservation of native and indigenous livestock and promoting sustainable production and (iv) linking the expenditures to national and international biodiversity targets as outlined by [9].

### BIOFIN methodological framework

We used the BIOFIN framework to analyse the mechanism of investment by the public sector in the animal husbandry department for the conservation of native and indigenous livestock diversity in Madhya Pradesh, India. We used the three stages of the BIOFIN framework (i) Policy and Institutional Review (PIR), (ii) Biodiversity Expenditure Review (BER), and (iii) Biodiversity Finance Needs Assessment (FNA) to identify the financial gaps and make the finance plans for conservation of native and indigenous livestock in Madhya Pradesh (Fig 1).

   **1. Policy and institutional review (PIR).** We used the data and schemes provided by the Animal Husbandry Department of Madhya Pradesh which focus on conserving native and indigenous livestock of the state (Table 1). The schemes implemented by the Department of Animal Husbandry, Madhya Pradesh, target a livestock population comprising 1.87 crore cattle including approximately 1.70 million indigenous breeds and 1.03 crore buffaloes [34]. We analyzed the financial schemes and activities associated with biodiversity expenditure and determined the relevant expenses for each of these multi-objective composite schemes. These schemes are in line with the BIOFIN categories [9].

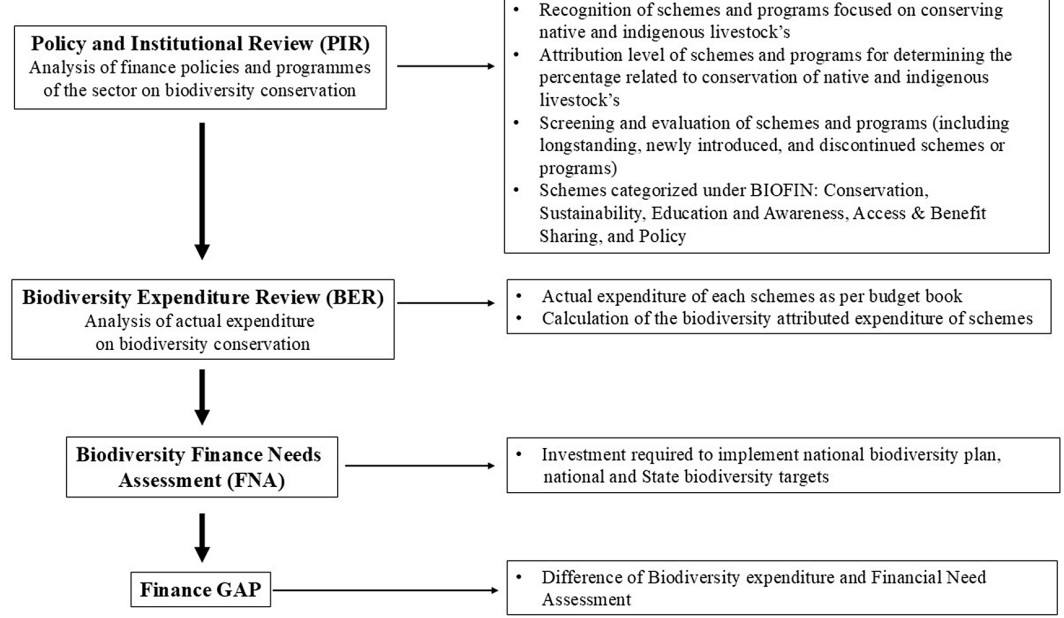

**Fig 1. Biodiversity Finance Initiative (BIOFIN) framework (BIOFIN, 2018) for the creation of Biodiversity Finance Plans for conservation of native and indigenous livestock in Madhya Pradesh, India.**

**Table 1. Attribution methodology for determining the percentage of identified schemes and programs related to the conservation of native and indigenous livestock in Madhya Pradesh, India. The Convention on Biological Diversity (CBD) has three main objectives: (i) the conservation of biological diversity, (ii) the sustainable use of the components of biological diversity, and (iii) the equitable sharing of the benefits arising out of the utilization of genetic resources).**

| Expenditure relevance to the conservation of native and indigenous livestock | Criteria | Expenditure attributable to biodiversity expenditure (% Range) | Level of biodiversity attribution of the scheme |
|---|---|---|---|
| Direct | The primary goal of the organization/activity is to achieve one of the three objectives outlined by the CBD. | 100−90% | 95% |
| Indirect Very High | The primary aim is to achieve at least one of the CBD objectives, along with other related and supportive goals. | 90−75% | 82.5% |
| Indirect High | The primary aim is to achieve at least one of the CBD objectives, with additional related and supportive goals to a lesser extent | 75−50% | 62.5 |
| Indirect Medium | Achieving at least one of the CBD Objectives or NBTs, alongside other non-biodiversity related goals or actions in a balanced manner | 50−25% | 37.5 |
| Indirect Low | The primary focus is on non-biodiversity activities, but there is an explicit aim for positive biodiversity impacts | 25−5% | 15% |
| Indirect Marginal | Minor biodiversity impacts are anticipated from larger non-biodiversity programs, provided that safeguards are in place | 5−0% | 2.5% |

Upon review, we categorized and mapped all schemes and programs into five BIOFIN categories [9]: (i) Conservation (for schemes addressing core conservation issues), (ii) Sustainable use (for schemes supporting breed conservation efforts), (iii) Education and awareness, (iv) Policy (for schemes aiding the planning process), and (v) Access & Benefit Sharing (for schemes promoting benefit sharing after the commercial use of biological resources). We gathered secondary data from the animal husbandry department by reviewing budget documents, annual reports, official websites, and detailed demand/

grant records and conducting consultations with the department for the financial years 2016−2017–2021−2022 to identify biodiversity-related schemes. We tracked the biodiversity-relevant schemes over the financial years from 2016−17–2021−22 using BIOFIN categories to examine the presence of these schemes and the trend in expenditure for biodiversity conservation. We used the secondary data of expenditure which was collected from the Detailed Demand for Grant Documents (DDG) or the budget book of the state's animal husbandry department for the period 2016−17–2021−22. We compiled the biodiversity investment data into Indian Rupees (INR).

**2. Biodiversity expenditure review (BER).** The BER process was conducted to determine the exact funding estimates for conserving the native and indigenous livestock diversity in Madhya Pradesh. This assessment was based on tracking fund flow and attributing biodiversity-related portions of the schemes. Since not all expenditure under a scheme was dedicated to conserving native and indigenous livestock, it was necessary to calculate the 'attributable share' for the conservation of native and indigenous livestock for each scheme individually [41]. We used the BIOFIN methodology to determine the 'attributable share,' along with its focus areas, targets, and scheme components, which was applied in consultation with the animal husbandry department to determine the expenditure attributable to native and indigenous livestock conservation. The approach identifies biodiversity investments and examines the budget to categorize items related to biodiversity. Attribution followed modified RIO Markers and was categorized into 'direct' and 'indirect' based on their 'attributable share' for native and indigenous livestock conservation. Budget lines that are directly and primarily focused on biodiversity conservation are assigned 100% attribution. The 'indirect' expenditures were further classified into Indirect Very High, Indirect High, Indirect Medium, Indirect Low, and Indirect Marginal to reflect varying levels of contribution according to specific objectives and mandates aligned with CBD objectives (see Table 1). Specific guidelines for schemes and programs were utilized to determine their direct or indirect contribution to native and indigenous livestock conservation. The actual expenditure for each identified scheme was compiled from budget documents. The biodiversity-attributed financial expenditure (biodiversity finance) was calculated based on the actual expenditure incurred during the year for each scheme or program.

**3. Financial need assessment (FNA).** The FNA was conducted as part of the MPBSAP (2018–2030) to estimate the financial resources required to meet the SBTs. The financial resource gap (difference between FNA and BER) is calculated using data from the Financial Need Assessment (FNA). This gap is assessed across specific sectors and the schemes required for SBTs. Although this financial gap is not useful for setting priorities for planning activities, understanding the magnitude of the resource gap and the biodiversity relevance of various schemes and programs assists in identifying suitable finance solutions and formulating the Biodiversity Finance Plan (BFP).

## Results

### Policy and institutional review (PIR)

A total of 67 schemes and programs were identified under the BIOFIN categories (conservation, sustainability, and education and awareness) for native and indigenous livestock conservation and sustainable production that were floated by the Animal Husbandry Department of Madhya Pradesh, India, during 2016–2017 (S1 Appendix). However, no schemes were identified for policy and Access & Benefit Sharing (ABS) categories during this period. Between 2016 and 2022, 24 schemes were discontinued, and a total of 43 schemes were functional in 2021–2022 (Table 2). Among these 43 schemes, 12 were focused on conservation, 27 on sustainability, and 4 aimed at education and awareness. During this period, the schemes related to conservation and education awareness decreased by 20% and 56%, respectively, while schemes related to sustainability increased by 80% (S1 Table).

In 2021–2022, the 43 active schemes were evaluated based on biodiversity attribution levels and grouped into direct and indirect categories. A significant majority, 90.7%, fell under the indirect category, while only 9.3% were classified as direct (S2 Table). Comparatively, in 2016–2017, 39 schemes were analyzed, with 23.07% categorized as direct and 76.93% as indirect (S2 Table).

Table 2. The Gross State Domestic Product (GSDP) and Biodiversity expenditure under BIOFIN schemes for the conservation of native and indigenous livestock during 2016 to 2022 in Madhya Pradesh, India.

| Financial Year | Madhya Pradesh Gross State Domestic Product (GSDP) | | Biodiversity attributed expenditure under BIOFIN schemes | | |
|---|---|---|---|---|---|
| | *INR (Lakh) | INR Cr. | INR (Source S1 Appendix) | INR Cr. | Percent of GSDP |
| 2016−17 | 649823 | 64982.30 | 2752326275 | 275.23 | 0.04 |
| 2017−18 | 726284 | 72628.40 | 2957802175 | 295.78 | 0.04 |
| 2018−19 | 829805 | 82980.50 | 3247903100 | 324.79 | 0.04 |
| 2019−20 | 927855 | 92785.50 | 4135591800 | 413.56 | 0.04 |
| 2020−21 | 961643 | 96164.30 | 4786333975 | 478.63 | 0.05 |
| 2021−22 | 1136137 | 113613.70 | 6465751700 | 646.58 | 0.06 |
| Total | 5231547 | 523154.70 | 24345709025 | 2434.57 | 0.05 |

Source* https://msmeindore.nic.in/State%20Profile-%20Madhya%20Pradesh.pdf

## Biodiversity expenditure review (BER)

A total of INR 5,159.88 crore (US $727.4 million) was spent between 2016–2022 on livestock conservation and sustainable production in Madhya Pradesh (Fig 2). Of this, 47.18% (approximately INR 2434.57 crore) (Fig 3), categorized by the level of biodiversity attribution (Table 1), within various schemes, was allocated either directly or indirectly (S2 Table) towards the conservation of native and indigenous livestock. This averages to an annual expenditure of INR 405.761 Cr. (US $286.42 million) and represents 0.05% of the total Gross State Domestic Product (GSDP) of Madhya Pradesh (Table 2). The net expenditure of the animal husbandry department on various conservation-related schemes increased from 39.5% of the total spending in 2016–2017 to 70.7% in 2021–2022 (Fig 2). The overall expenditure attributed to biodiversity reported a substantial increase of 57.8% in public investments (INR 275.23 Cr. to 646.58 Cr.) during 2016–2017–2021–2022 (Fig 3). The conservation-based schemes were received the most financial attention and reported increased during this period (67.02 to 95.3%) while funding investments decreased for sustainability (26.5 to 3.5%), education and awareness (9.4 to 1.2%) schemes (Fig 3). Based on level of biodiversity attribution, the direct attribution (95% attribution) accounted for 5.8 to 28.77% of the expenditure, while indirect attribution with very high attribution (82.5%) accounted 40.13 to 73.45%, this is higher than others (Fig 4).

## Financial need assessment (FNA) and financial gap

The FNA was conducted for the period 2019–2022 to estimate the funding required to meet SBTs outlined in the Madhya Pradesh Biodiversity Strategy and Action Plan (MPBSAP), 2018–2030. The assessment compared the estimated financial needs with actual expenditures identified through BER.

In 2021–2022, the actual expenditure exceeded the estimated financial need by INR 117.26 crore (approximately $13.8 million), indicating a surplus in biodiversity-related funding for that year. However, in the previous years, 2019–2020 and 2020–2021, there were funding shortfalls of INR 23.89 crore and INR 2.57 crore, respectively (Table 3).

Despite the surplus in 2021–2022, the analysis highlights the need for continued investment and the development of new schemes to fully meet the NBTs and SBTs. (Table 4). The financial gap analysis provides valuable insights into the alignment between funding availability and biodiversity conservation goals. It also helps identify areas where additional resources or policy adjustments may be necessary.

## Discussion and conclusions

To achieve India's 12 National Biodiversity Targets (NBTs), the Government of India estimated an annual biodiversity expenditure of approximately US$ 2.64 billion over a five-year period [41]. However, sub-national or state-level data on

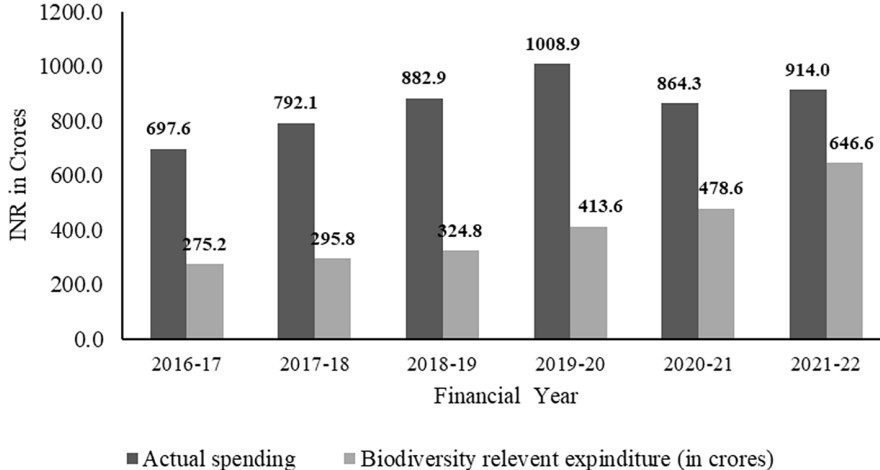

**Fig 2. Analysis of appropriation accounts for the existing schemes and programs for the conservation of native and indigenous livestock during 2016 to 2022 in Madhya Pradesh, India.**

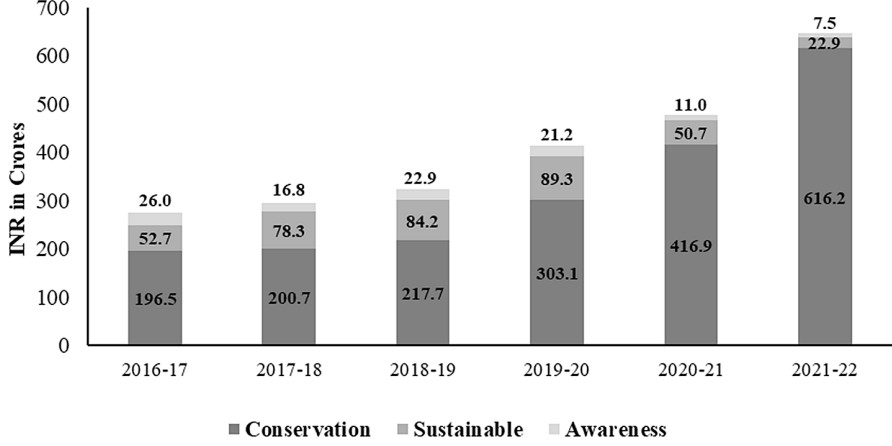

**Fig 3. Biodiversity expenditure (INR 2434.57 Cr.)under BIOFIN schemes for the conservation of native and indigenous livestock during 2016 to 2022 in Madhya Pradesh, India.**

biodiversity investments in India remain limited. For example, the Indian state of Punjab's biodiversity-related expenditure was estimated at INR 124 crore [7], but this figure encompasses all biodiversity sectors and lacks detailed breakdowns by sector. In particular, there is a significant knowledge gap regarding biodiversity investments in the animal husbandry sector, which plays a vital role in India's rural economy. This study provides the first public estimate of biodiversity investments specifically focused on conserving native and indigenous livestock in Madhya Pradesh. The analysis is based on primary data from state budget documents and employs the standardised BIOFIN methodology, enabling consistent tracking and comparison of biodiversity-related expenditures.

We found that biodiversity-attributed investments in Madhya Pradesh have increased steadily for the conservation of native and indigenous livestock to about INR 646.58 crore in 2021–2022, which is about 0.06% of the state's gross domestic product (GSDP). While this is a positive trend, it remains below the IUCN's recommendation for OECD countries

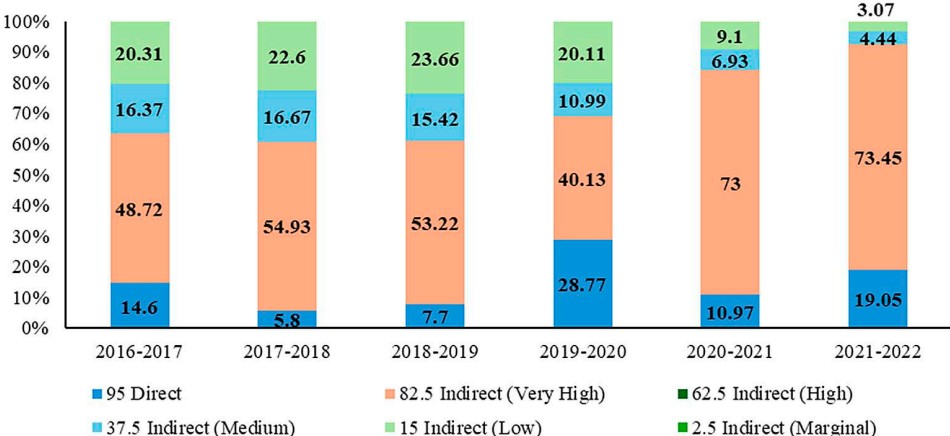

**Fig 4. Biodiversity attributed expenditure (percentage) in different schemes for the conservation of native and indigenous livestock during 2016 to 2022 in Madhya Pradesh, India.**

**Table 3. Financial need assessment and Biodiversity Expenditure Review (INR Cr.) for the conservation of native and indigenous livestock from 2019−20 to 2021−22 in Madhya Pradesh, India.**

| Particular | 2019−20 | 2020−21 | 2021−22 |
|---|---|---|---|
| Financial Need Assessment (FNA) (As per MPBSAP-2018-30) in INR Cr. | 437.45 | 481.2 | 529.32 |
| Biodiversity Expenditure Review (BER) in INR Cr. | 413.56 | 478.63 | 646.58 |
| Financial Gap -Difference between BER to FNA in INR Cr. | −23.89 | −2.57 | 117.26 |

to allocate at least 0.3% of GDP to biodiversity conservation. [40]. Globally, public biodiversity investments have increased steadily to an average of about 0.19–0.25% of global GDP [42]. Thus, Madhya Pradesh's current spending, at 0.05% of GSDP, suggests room for growth, especially when considering other biodiversity components like forests and agriculture.

Over a six-year period analysed, the average biodiversity-attributed expenditure was 14.5% for direct investments, while indirect investments made up to 85.5%. These attributions were determined based on expert consultations and stakeholder input, assessing the degree to which these budget items contribute to biodiversity outcomes. Our study comprehensively collected and analysed the primary biodiversity investment data for the first time by employing a standard methodology comparable to other data.

Although the financial needs assessment revealed a funding surplus in 2021–2022, the overall trend indicates a need for continued resource mobilization. The PIR shows that 12 new schemes were launched in 2021−22 to address the native and indigenous livestock conservation issues, which included initiatives such as Livestock Breeding Farms, Gau Samvardhan, Livestock Sanctuaries, National Livestock Mission, and the Intensive Livestock Development Programme. Additionally, Vidyasagar Gau Samvardhan, Poultry Farms for breed conservation. These schemes have worked well for the conservation of native and indigenous livestock breeds. Some of the successful breeds include Murrah buffalo, Gir, Sahiwal, Rathi, Red Sindhi cattle, Ongole cattle, Punganur cattle, Malnad Gidda cattle, Deoni cattle, Hallikar cattle, and Kadaknath poultry. Sheep Farms for sheep conservation, and the Rashtriya Krishi Vikas Yojna for nucleus herd development and the promotion of indigenous or native breed sorted semen were purposefully adopted to address conservation issues. However, the breeding strategy implemented included the induction of indigenous breeds, conservation efforts on breeding farms, restriction of crossbreeding programs within breeding tracts, utilization of frozen semen from native and indigenous breeds, and distribution of indigenous and native breed livestock. Both the natural and artificial insemination

**Table 4. Gaps analysis in the existing schemes and programs for the conservation of native and indigenous livestock during 2016 to 2022 in Madhya Pradesh, India.**

| Category | Issues- gaps |
|---|---|
| Conservation | 1. Regular restructuring of the budget demand and allocation |
| | 2. Programme for organic Farming Natural/organic farming |
| | 3. Specific programme for the management of breeding tact of native livestock. |
| | 4. Policy to declare a Biodiversity Heritage Site for native breeds. |
| Sustainable | Breed conservation programme through incentive to farmers. |
| | Programmes for Climate Change adaptation strategy |
| | Feed and fodder production programme |
| Education and Awareness | Promote Livestock conservation-based Biodiversity Education and awareness programme |
| Planning | Policies and regulations to conserve and sustain the native livestock. Review of Breeding policy and implementation |
| Access to Benefit Sharing (ABS) | Survey and identification of tradable biological resources with documentation of Traditional Knowledge |

technology were used for successful breeding purposes. It includes the semen of specific native and indigenous breeds of cattle and buffalo were used in the respective breeds in their native tracts in the state (i.e., Tharparkar in Rajasthan, Gir in Gujarat). Despite these efforts, several key conservation schemes, such as calf rearing programs for cows and buffaloes, the National Kamdhenu Breeding Centre (NKBC), Embryo Transfer Technology (ETT), and the Animal Breeding Centre, were discontinued. Similarly, in the sustainability category, while 25 new schemes were launched, important programs like the National Animal Health & Disease Prevention and Fodder Production schemes were phased out. Among the 25 new schemes, notable ones were the Division and District-level Animal Husbandry and Dairy initiatives, the Strengthening of Veterinary Hospitals and Dispensaries, the establishment of the Veterinary Council of India, Mobile Veterinary Unit Central, and the establishment of new Veterinary Hospitals and Dispensaries. Additionally, the Strengthening of Veterinary Hospitals, Mobile Veterinary Services, Veterinary Hospital Infrastructure Development, and other related programs have been introduced. In the awareness category, seven schemes were assessed to focus on education and awareness, but the closure of important schemes such as the Animal Exhibition and Gopal Puraskar Yojna highlights a gap in public engagement. The reasons of discontinued conservation schemes possibly (i) achievements of the scheme's objectives and targets, (ii) financial constraints (discontinued due to budget limitations or reallocation of funds) to other priorities in the livestock sector for the other conservation schemes), (iii) shifting priorities as per the state policy according to relevance.

The BIOFIN methodology has proved effective in developing strong finance plans for the sector by addressing issues related to conservation, sustainability, and benefit-sharing. In Madhya Pradesh, biodiversity targets have been outlined in the MPBSAP, 2018-30 [43]. The Animal Husbandry Department of Madhya Pradesh is utilizing its budget to meet these targets by implementing schemes focused on conserving native livestock. However, to ensure long-term sustainability, it is essential to develop integrated policies that link livestock management with agricultural ecosystems, food security, disease control, and rural livelihoods.

Local breeds are critical to rural communities but are often neglected in policy and investment decisions [44]. Therefore, the conservation regulations and state breeding policies need to be revisited. The sector has increased production by introducing exotic germplasm to local breeds, but it has also posed significant threat to native livestock diversity. The Animal Husbandry Department has responded by investing in conservation, disease control, research, infrastructure, and market development. Nonetheless, challenges such as declining agricultural roles for animals, low productivity of native breeds, habitat changes, and limited feed and fodder availability persist. To address these issues, investment restructuring and new schemes are needed that will enhance breed productivity, support feed and fodder production, and strengthen rural livelihoods.

Madhya Pradesh has made commendable progress in implementing its State Biodiversity Action Plan and state biodiversity targets [43]. As one of India's states rich in indigenous and livestock diversity, it has worked towards achieving Aichi Target 13 and National Biodiversity Target 07, which focus on protecting genetic diversity in socio-economically and culturally significant species, including native and indigenous livestock. Additionally, it aligns with Target 4 of the Kunming-Montreal Global Biodiversity Framework (2022) to halt species extinction and protect genetic diversity. However, schemes related to Aichi Target 1, aimed at fostering biodiversity awareness, are notably absent. This corresponds to the gap in implementing National Biodiversity Target 1 and Target 20 of the Kunming-Montreal Global Biodiversity Framework, which emphasizes strengthening capacity-building, technology transfer, and scientific and technical cooperation for biodiversity.

This study presents a unique approach by identifying various schemes that contribute, directly or indirectly, to the conservation of native and indigenous livestock and mapping their associated expenditures. Such an approach should be extended to other states in India. By highlighting gaps in financial planning and the implementation of schemes for conserving native and indigenous livestock, the study suggests that similar financial challenges may also affect other public sectors. Consequently, this approach can serve as a replicable model for other states as well as other developing countries.

## Supporting information

**S1 Table. Status of schemes in the animal husbandry department for the conservation of native and indigenous livestock in Madhya Pradesh, India during 2016–2022.**
(DOCX)

**S2 Table. The number of schemes with Biodiversity Attributed to conserving native and indigenous livestock in the animal husbandry department of Madhya Pradesh, India from 2016 to 2022.**
(DOCX)

**S1 Appendix. Data of various programs and schemes run by the Animal Husbandry Department in Madhya Pradesh during 2016-2022.**
(XLSX)

## Acknowledgments

We are grateful to our Chairperson, Dr. D.K. Bandyopadhyay, as well as the external expert members, Dr. Dipankar Ghose and Dr. Rahul Kaul, of the Departmental Research Committee at the Amity Institute of Forestry and Wildlife, Amity University Uttar Pradesh, for their valuable suggestions and comments, which helped refine the study design and facilitate the successful execution of this research.

## Author contributions

**Conceptualization:** Bakul Lad, Randeep Singh.

**Data curation:** Bakul Lad.

**Formal analysis:** Bakul Lad, Faiyaz A Khudsar, Randeep Singh.

**Investigation:** Bakul Lad, Faiyaz A Khudsar, Randeep Singh.

**Methodology:** Bakul Lad, Ajay Sharma, Randeep Singh.

**Project administration:** Bakul Lad.

**Resources:** Bakul Lad.

**Supervision:** Faiyaz A Khudsar, Ajay Sharma, Randeep Singh.

**Validation:** Bakul Lad, Randeep Singh.

**Writing – original draft:** Bakul Lad, Randeep Singh.

**Writing – review & editing:** Bakul Lad, Faiyaz A Khudsar, Ajay Sharma, Randeep Singh.

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
