## [Decision Letter · Decision Letter 0]

26 May 2025

PONE-D-25-17885Finance for Sustainability: Exploring the BIOFIN Framework for Government Investment in Conserving Native and Indigenous Livestock Breeds in Central IndiaPLOS ONE

Dear Dr. Singh,

Thank you for submitting your manuscript to PLOS ONE. After careful consideration, we feel that it has merit but does not fully meet PLOS ONE’s publication criteria as it currently stands. Therefore, we invite you to submit a revised version of the manuscript that addresses the points raised during the review process.

We look forward to receiving your revised manuscript.

Kind regards,

Kasi Eswarappa

Academic Editor

PLOS ONE

Journal Requirements:

2. We note that Figure 1 in your submission contain [map/satellite] images which may be copyrighted. All PLOS content is published under the Creative Commons Attribution License (CC BY 4.0), which means that the manuscript, images, and Supporting Information files will be freely available online, and any third party is permitted to access, download, copy, distribute, and use these materials in any way, even commercially, with proper attribution. For these reasons, we cannot publish previously copyrighted maps or satellite images created using proprietary data, such as Google software (Google Maps, Street View, and Earth). For more information, see our copyright guidelines: http://journals.plos.org/plosone/s/licenses-and-copyright.

Natural Earth (public domain): http://www.naturalearthdata.com

Additional Editor Comments:

I appreciate your effort and recommend to carry major revisions to the manuscript.

Reviewers' comments:

Reviewer's Responses to Questions

**Comments to the Author**

1. Is the manuscript technically sound, and do the data support the conclusions?

Reviewer #1: Yes

Reviewer #2: Partly

2. Has the statistical analysis been performed appropriately and rigorously? 

Reviewer #1: N/A

Reviewer #2: Yes

3. Have the authors made all data underlying the findings in their manuscript fully available?

Reviewer #1: Yes

Reviewer #2: Yes

4. Is the manuscript presented in an intelligible fashion and written in standard English?

Reviewer #1: Yes

Reviewer #2: Yes

5. Review Comments to the Author

Reviewer #1: The manuscript entitled “Finance for Sustainability: Exploring the BIOFIN Framework for Government Investment in Conserving Native and Indigenous Livestock Breeds in Central India” by Lad et al is a technically correct research paper with quality data that supports the conclusions. This article addresses a critical aspect of biodiversity conservation, financial resources allocated to protecting native and indigenous livestock. By examining the case of Madhya Pradesh, India, and employing the BIOFIN approach, the authors provide a unique and insightful analysis of how public sector investments are directed towards achieving Aichi Biodiversity Targets and sustainable development goals within the animal husbandry sector. The identification and categorisation of 43 relevant schemes, along with the quantification of expenditures, offer a clear and systematic overview of the financial landscape. The study's key contribution lies in demonstrating a replicable methodology for assessing financial availability and alignment with biodiversity targets. The conclusion that the Madhya Pradesh sector is meeting its assessed financial needs through systematic investments is encouraging. However, the paper also judiciously points out the inherent tension between increasing native livestock production and maintaining overall livestock biodiversity, a crucial consideration for long-term conservation strategies. The call to extend this approach to other states and developing countries underscores the broader significance of this research for enhancing financial planning and implementation in biodiversity conservation across various sectors. Using the Biodiversity Finance Initiative (BIOFIN) approach in the context of livestock biodiversity is a significant and valuable aspect of the study. The mapping and categorisation of numerous schemes provide a structured and comprehensive understanding of financial flows. The reporting of the total expenditure on livestock conservation offers concrete evidence of financial commitment. The study clearly articulates the potential for its methodology to be adopted by other regions facing similar challenges. The findings offer valuable insights for policymakers and practitioners involved in biodiversity conservation and financial planning. Overall, the article is well-written, informative, and communicates the key findings and implications. The manuscript presented in an intelligible fashion, with structured statistical analysis and written in standard English, but needs some language correction and recommendations for publication.

Reviewer #2: Finance for Sustainability: Exploring the BIOFIN Framework for Government Investment in Conserving Native and Indigenous Livestock Breeds in Central India

Comments and Suggestions

Line 63-66: mention some countries and need citation.

Line 78: delete the phrase “natural resource”

Line 113-114: what do you mean changes in land use

What are your criteria to say native or indigenous?

Line 119: Capra aegagrus hircus?

Line 126: 7,48,92,284.68?

What is the effect or output from each scheme? e.g Make a comparison in population and productivity trend for conserved breed between initial year and current year.

how many skilled personnel were produced from awareness scheme?

Which strategy or approach is used for conservation?

What type of breeding strategy implemented or implementing?

If there is incentive for participant farmers; what type of incentive,

What is the criteria for budget allocation for each scheme?

Explain in detail for the sustainable issues (discontinued projects such as: calf rearing for cows and buffaloes, the National Kamdhenu Breeding Centre (NKBC), Embryo Transfer Technology (ETT), and the Animal Breeding Centre were discontinued. In the sustainable category, 27 schemes were evaluated to address sustainable issues, but key schemes like National Animal Health & Disease Prevention and Fodder Production were discontinued.) and others.

6. PLOS authors have the option to publish the peer review history of their article (what does this mean? ). If published, this will include your full peer review and any attached files.

**Do you want your identity to be public for this peer review?** For information about this choice, including consent withdrawal, please see our Privacy Policy .

Reviewer #1: **Yes: ** Dr. Pranab Jyoti Das, Principal Scientist, ICAR-NRC on Pig

Reviewer #2: **Yes: ** Ebadu Areb

---

## [Author Response · Author response to Decision Letter 1]

29 May 2025

Financing Sustainability: Applying the BIOFIN Framework to Government Investments in Conserving Native and Indigenous Livestock Breeds in Central India.

Reviewer #1:

Comments 1: Overall, the article is well-written, informative, and communicates the key findings and implications. The manuscript presented in an intelligible fashion, with structured statistical analysis and written in standard English, but needs some language correction and recommendations for publication.

Our response: We are grateful to the reviewer for taking his valuable time to look at our work. We thoroughly reviewed the manuscript and made significant language and grammar edits throughout that further improved its readability.

Reviewer #2:

Comments 1: Line 63-66: mention some countries and need citation.

Our response: We have now added a list of the countries and the associated citations in the revised manuscript. Please see lines 62 to 65. Thank you.

Comments 2: Line 78: delete the phrase “natural resource”

Our response: Suggestion accepted. We have removed the phrase.

Comments 3: Line 113-114: what do you mean changes in land use

Our response: We have removed the phrase.

Comments 4: What are your criteria to say native or indigenous?

Our response: We have now added a paragraph (lines 119-121) to explain the criteria we used to classify a species into native or indigenous. Accordingly, both native and indigenous breeds originated in India. However, native breeds are specific to Madhya Pradesh (our study area), while indigenous breeds may also come from other parts of India.

Comments 5: Line 119: Capra aegagrus hircus?

Our response: We removed the word aegagrus.

Comments 6: Line 126: 7,48,92,284.68?

Our response: We understand this way of representing large numbers could be confusing to readers. Therefore, we have now written this number as 74.9 million Indian rupees. Line number 129.

Comments 6: What is the effect or output from each scheme? e.g Make a comparison in population and productivity trend for conserved breed between initial year and current year.

Our response: Although the impact assessment of each scheme is an important question, it was beyond the scope of our study. Our study objectives focused solely on studying the financial investment for native breed conservation using the Biofin Methodology.

Comments 7: Which strategy or approach is used for conservation?

Our response: The conservation criteria—effects and outcomes—had already been addressed by the Madhya Pradesh Animal Husbandry Department, so our study did not include them.

Comments 7: how many skilled personnel were produced from awareness scheme?

Our response: The awareness schemes were constituted to aware the farmers and community only to promote the rearing of native and indigenous livestock. The awareness programme does not cover the skill development.

Comments 8: What type of breeding strategy implemented or implementing?

Our response: We have incorporated this information in line numbers 350 to 360. The Policy and Institutional Review highlights various strategies and approaches for conservation, including Livestock Breeding Farms, Gau Samvardhan, Livestock Sanctuaries, the National Livestock Mission, the Intensive Livestock Development Programme, and the induction of both small and large animals. Additionally, Vidyasagar Gau Samvardhan, Poultry Farms for breed conservation, Sheep Farms for sheep conservation, and the Rashtriya Krishi Vikas Yojna for nucleus herd development and the promotion of indigenous or native breed sorted semen were purposefully adopted to address conservation challenges for making strategies.

Comments 9: If there is incentive for participant farmers; what type of incentive,

Our response: There are no such direct incentives for farmers. However, continuous awareness and consecutive training may be considered as incentive.

Comments 10: What are the criteria for budget allocation for each scheme?

Our response: The criterion of budget allocation depended on the requirement of the fund proposed by the department as per the requirement to fulfil the objectives of programmes and schemes.

Comments 11: Explain in detail for the sustainable issues (discontinued projects such as: calf rearing for cows and buffaloes, the National Kamdhenu Breeding Centre (NKBC), Embryo Transfer Technology (ETT), and the Animal Breeding Centre were discontinued. In the sustainable category, 27 schemes were evaluated to address sustainable issues, but key schemes like National Animal Health & Disease Prevention and Fodder Production were discontinued.) and others.

Our response: We have incorporated this information in line number 365 to 371. The sustainable issues for the discontinued schemes, the 25 New schemes were new launched according to the policy of the government to fulfil the gaps of the discontinued schemes. Among them the schemes like Division and District level AH and Dairy, Strengthening of VET Hospitals and dispensaries, Est. Veterinary Council of India, Mob VET Unit Central, Est of New Vet Hospital and Dispensary Strengthening of VET Hospitals Mob VET Services, Vet Hosp and Other Infra Development, etc have been newly launched.

---

## [Decision Letter · Decision Letter 1]

24 Jul 2025

PONE-D-25-17885R1Financing Sustainability: Applying the BIOFIN Framework to Government Investments in Conserving Native and Indigenous Livestock Breeds in Central IndiaPLOS ONE

Dear Dr. Singh,

Thank you for submitting your manuscript to PLOS ONE. After careful consideration, we feel that it has merit but does not fully meet PLOS ONE’s publication criteria as it currently stands. Therefore, we invite you to submit a revised version of the manuscript that addresses the points raised during the review process.

We look forward to receiving your revised manuscript.

Kind regards,

Kasi Eswarappa

Academic Editor

PLOS ONE

Journal Requirements:

Additional Editor Comments:

I advise you to check the comments thoroughly and address it in detail without leaving for further editing.

Reviewers' comments:

Reviewer's Responses to Questions

**Comments to the Author**

1. If the authors have adequately addressed your comments raised in a previous round of review and you feel that this manuscript is now acceptable for publication, you may indicate that here to bypass the “Comments to the Author” section, enter your conflict of interest statement in the “Confidential to Editor” section, and submit your "Accept" recommendation.

Reviewer #1: All comments have been addressed

Reviewer #2: All comments have been addressed

2. Is the manuscript technically sound, and do the data support the conclusions?

Reviewer #1: Yes

Reviewer #2: Partly

3. Has the statistical analysis been performed appropriately and rigorously? 

Reviewer #1: Yes

Reviewer #2: Yes

4. Have the authors made all data underlying the findings in their manuscript fully available?

Reviewer #1: Yes

Reviewer #2: Yes

5. Is the manuscript presented in an intelligible fashion and written in standard English?

Reviewer #1: Yes

Reviewer #2: Yes

6. Review Comments to the Author

Reviewer #1: This manuscript by Lad et al. is a commendable and technically sound research paper that makes a valuable and timely contribution to biodiversity conservation. Its innovative application of the BIOFIN framework to analyze government investment in conserving native and indigenous livestock breeds is particularly impressive, offering a robust methodology. The detailed financial analysis, clear insights into policy implications, and recognition of complex conservation trade-offs make this work impactful. This well-written and informative study is a significant advancement in the field and is recommended for acceptance.

Reviewer #2: The authors try to address the previous comments. However, still there are some concerns;

1. Line 123-129: Need appropriate citation for listed native and indigenous breeds.

2. From a total of identified 43 schemes from 2016 to 2022, how many of them are successful? How many discontinued? For which breed of conservation is more successful?

3. If conservation criteria, effects and outcomes had already been addressed by the Madhya Pradesh Animal Husbandry Department, include the information in the background (introduction) section.

4. To get good information about the conservation program and to understand BIOFIN for your target population: you should include information such as number of participant farmers or experts, number of animals for each breed, number of farms……etc in table form in the methodology section.

5. Does the breeding strategy is natural or artificial? using sorted semen for which breed? What is the distribution modality of native and indigenous breeds?

6. What is the reason for discontinued of some conservation schemes?

7. Use consistence reference format

7. PLOS authors have the option to publish the peer review history of their article (what does this mean? ). If published, this will include your full peer review and any attached files.

**Do you want your identity to be public for this peer review?** For information about this choice, including consent withdrawal, please see our Privacy Policy .

Reviewer #1: **Yes: ** Dr. Pranab Jyoti Das, Principal Scientist , Indian Council of Agricultural Research

Reviewer #2: **Yes: ** Ebadu Areb

---

## [Author Response · Author response to Decision Letter 2]

29 Jul 2025

Financing Sustainability: Applying the BIOFIN Framework to Government Investments in Conserving Native and Indigenous Livestock Breeds in Central India.

Reviewer #1:

Comments 1: Line 123-129: Need appropriate citation for listed native and indigenous breeds.

Our response: We are grateful to the reviewer for taking his valuable time to look at our work. We have now added the associated citations in the revised manuscript. Please see lines 123 to 129. Thank you

Comments 2: From a total of identified 67 schemes from 2016 to 2022, how many of them are successful? How many discontinued? For which breed of conservation is more successful?

Our response: Since 2016, a total of 43 schemes have continued successfully, while 24 were discontinued. Please see lines 252 to 255. These schemes have worked well for the conservation of native and indigenous livestock breeds. Some of the successful breeds include Murrah buffalo, Gir, Sahiwal, Rathi, Red Sindhi cattle, Ongole cattle, Punganur cattle, Malnad Gidda cattle, Deoni cattle, Hallikar cattle, and Kadaknath for poultry conservation. We have included in line number 356 to 359.

Comments 3: If conservation criteria, effects and outcomes had already been addressed by the Madhya Pradesh Animal Husbandry Department, include the information in the background (introduction) section.

Our response: Suggestion accepted. We have included in the background. We have included in line number 133 to 137. The Animal Husbandry Department has assessed the impacts and outcomes of its various initiatives, with a particular focus on disease control, infrastructure development, and the enhancement of animal welfare and productivity. Additionally, the department has advanced breeding programmes and genetic improvement initiatives, complemented by targeted training and awareness campaigns.

Comments 4: To get good information about the conservation program and to understand BIOFIN for your target population: you should include information such as number of participant farmers or experts, number of animals for each breed, number of farms……etc in table form in the methodology section

Our response: The objectives of the study are to find out the trend of financing for implemented schemes by department of animal husbandry for using BIOFIN methodology conservation of the indigenous/ native breeds of livestock. The issue does not relate with the objectives of the study therefore the no of farmers or experts not included. As per suggestion In Methodology section livestock population included. The target population for the schemes implemented by the Department of Animal Husbandry Madhya Pradesh is the 1.87 crore (18.8 million) (1.70 million indigenous population) cattle population and 1.03 crore the Buffalo population (19 Animal Census). We have included in line number 189 to 191.

Comments 5: Does the breeding strategy is natural or artificial? using sorted semen for which breed? What is the distribution modality of native and indigenous breeds??

Our response: As per the Policy and institutional review of the schemes, both the natural and artificial insemination technology were used for successful breeding purposes. It includes the semen of specific native and indigenous breeds of cattle and buffalo were used in the respective breeds in their native tracts in the state (i.e. Tharparkar in Rajasthan, Gir in Gujarat). We have included in line number 171 to 174.

Comments 6: What is the reason for discontinued of some conservation schemes

Our response: The reasons of discontinued conservation schemes are (i) achievements of the scheme's objectives and targets, (ii) financial constraints (discontinued due to budget limitations or reallocation of funds) to other priorities in the livestock sector for the other conservation schemes), (iii) shifting priorities as per the state policy according to relevance. We have included in line number 387 to 390.

Comments 7: Use consistence reference format

Our response: As per suggestion references format corrected.

---

## [Editor Report · Decision Letter 2]

6 Aug 2025

Financing Sustainability: Applying the BIOFIN Framework to Government Investments in Conserving Native and Indigenous Livestock Breeds in Central India

PONE-D-25-17885R2

Dear Dr. Randeep Singh,

We’re pleased to inform you that your manuscript has been judged scientifically suitable for publication and will be formally accepted for publication once it meets all outstanding technical requirements.

Kind regards,

Kasi Eswarappa

Academic Editor

PLOS ONE

Additional Editor Comments (optional):

Our best wishes to authors.
---

## [Editor Report · Acceptance letter]

PONE-D-25-17885R2

PLOS ONE

Dear Dr. Singh,

I'm pleased to inform you that your manuscript has been deemed suitable for publication in PLOS ONE. Congratulations! Your manuscript is now being handed over to our production team.

Kind regards,

on behalf of

Dr. Kasi Eswarappa

Academic Editor

PLOS ONE